# Research Progress and the Prospect of Damping Magnesium Alloys

**DOI:** 10.3390/ma17061285

**Published:** 2024-03-11

**Authors:** Jinxing Wang, Yi Zou, Cong Dang, Zhicheng Wan, Jingfeng Wang, Fusheng Pan

**Affiliations:** 1College of Materials Science and Engineering, Chongqing University, Chongqing 400030, China; 202209021040@stu.cqu.edu.cn (Y.Z.); dangcong_24@163.com (C.D.); 202209021059@stu.cqu.edu.cn (Z.W.); jfwang@cqu.edu.cn (J.W.); fspan@cqu.edu.cn (F.P.); 2National Engineering Research Center for Magnesium Alloys, Chongqing University, Chongqing 400030, China

**Keywords:** damping magnesium alloy, damping capacity, research progress

## Abstract

As the lightest structural metal material, magnesium alloys possess good casting properties, high electrical and thermal conductivity, high electromagnetic shielding, and excellent damping properties. With the increasing demand for lightweight, high-strength, and high-damping structural materials in aviation, automobiles, rail transit, and other industries with serious vibration and noise, damping magnesium alloy materials are becoming one of the important development directions of magnesium alloys. A comprehensive review of the progress in this field is conducive to the development of damping magnesium alloys. This review not only looks back on the traditional damping magnesium alloys represented by Mg-Zr alloys, Mg-Cu-Mn alloys, etc. but also introduces the new damping magnesium materials, such as magnesium matrix composites and porous magnesium. But up to now, there have still been some problems in the research of damping magnesium materials. The effect of spiral dislocation on damping is still unknown and needs to be studied; the contradiction between damping performance and mechanical properties still lacks a good balance method. In the future, the introduction of more diversified damping regulating methods, such as adding other elements and reinforcements, optimizing the manufacturing method of damping magnesium alloy, etc., to solve these issues, will be the development trend of damping magnesium materials.

## 1. Introduction

With the development of modern industries such as automobiles, electronics, aviation, etc., vibration and noise become issues that cannot be ignored [1,2]. The problems caused by vibration and noise are very difficult. The structure of products may malfunction under certain high-frequency random vibrations, and some may even lead to disasters [3,4]. By observing product failure reports, about two-thirds of industrial malfunctions are caused by vibration and noise. The development of materials that integrate high-damping and high-strength functional structures has become a hot research topic [5,6]. Damping alloys are a type of alloy that can meet the needs of high strength and high damping [7]. They have great commercial prospects and have been widely used in many fields [8,9]. But with the advancement of technology, the existing damping alloys are gradually unable to meet the increasingly demanding usage conditions. There is an urgent need to develop new damping alloys with better comprehensive performance. The density of pure magnesium is ρ = 1.738 g/cm^3^, approximately 2/3 of Al and 1/4 of steel [10,11,12,13]. Mg is currently one of the lightest metals in industrial applications [14] and can well meet the lightweight requirements of modern industrial machinery and equipment. Its crustal resources are abundant, making it a lightweight, energy-saving, and environmentally friendly material, often used in the fields of automobiles, aerospace, and electronic products [15]. At the same time, pure Mg has excellent damping capacity (Q^−1^ = 0.11, ε = 10^−4^), with a damping attenuation coefficient 5 times that of cast iron and 12 times that of aluminum alloys [16,17,18]. Its high damping advantage is particularly prominent among various metals.

To compare the ultimate tensile strength and damping performance of different metal damping alloys, the ultimate tensile strength R_m_ and SDC (specific damping capacity) values of some metals are shown in Figure 1. When the specific damping value is greater than 0.1 but less than 1, this type of alloy is defined as a low-damping metal. When the specific damping value is greater than 1 but less than 10, this type of alloy is defined as a medium-damping metal. When the specific damping value is greater than 10 but less than 100, this type of alloy is defined as a high-damping metal.

From Figure 1, the damping capacities of Mg-X alloys are usually relatively high. Most of them are greater than 10 but less than 100 and belong to high-damping alloys. In recent years, Mg-based damping alloys have received increasing attention due to their unique high-damping advantages. Their research is also in full swing. Meanwhile, magnesium also possesses unparalleled advantages over other alloys, such as excellent specific strength, specific stiffness, cutting performance, radiation resistance, high electromagnetic shielding, and many other advantages [19,20,21,22,23]. Therefore, the research and development of magnesium-based damping alloys is highly beneficial for this field. However, the mechanical properties of high-damping magnesium alloys are poor [24,25,26]. The tensile strength is only 100 MPa and the elastic modulus is 45 GPa [27], which is relatively low and makes it difficult to meet the usage conditions of structural materials [24]. For example, the yield strength of ZK01 alloys at room temperature is only 50–60 MPa [28], and the tensile strength of AZ91D alloys is only about 160 MPa [29]. Therefore, balancing the contradiction between the mechanical properties and damping properties of magnesium alloys becomes an important challenge in this field [30].

From Table 1, it can be seen that magnesium alloys have poorer mechanical properties compared to commonly used alloys, but their density, ductility, and specific stiffness are all good. Therefore, their application in damping materials is very promising.

This article aims to explore the research progress of damping magnesium alloys with integrated structures and functions and predict their development trends.

## 2. Damping Magnesium Alloy System

The metal element commonly used for alloying and adding to the magnesium matrix of damping magnesium alloys should meet one of the following two conditions [32]: one is that the solid solubility in α-Mg should be small, and the other is that it can form a eutectic structure with magnesium. The main alloying elements include Al, Zn, Li, Ag, Zr, Th, Mn, Ni, and rare earth (RE) elements. These elements possess functions such as solid solution strengthening, precipitation strengthening, and fine grain strengthening in magnesium alloys [33,34,35]. The role of alloying elements in magnesium alloys varies. Some elements, such as small amounts of Al, Mn, Zr, Zn, Be, etc., can improve the strength of magnesium alloys [36], while various alloying elements, such as Ce, Nd, Y, Si, Ca, Ti, B, Sr, Sb, Bi, Pr, etc., have been used to improve the working temperature and mechanical properties of modified AZ91 alloys [37]. Mn can also improve the corrosion resistance of alloys [38]. Zr can refine grains and improve the resistance to hot cracking [39]. Rare earth elements not only have similar effects to Zr but also improve casting performance, welding performance, and heat resistance and eliminate the stress corrosion tendency [40,41]. As an added element to damping alloys, it has different functions and can meet different needs.

Figure 2 shows the relationship between dislocation and damping consumption. Dislocation is a phenomenon in which one or several rows of atoms in a metal are arranged in a staggered manner. As a way of energy consumption, there are many dislocations in the crystal, which is the main reason why magnesium alloys have high damping performance. The density, length, and distribution of dislocations significantly affect the damping performance of magnesium alloys. The G-L dislocation theory [42,43] indicates that both strong and weak pinning points are pinning dislocations. When the strain amplitude is low, dislocation damping is resonant, and the internal friction is independent of the amplitude. Currently, vacancies and solute atoms are commonly used as weak pinning points in crystals which then pin dislocations [44]. Under external stress, there is a reciprocating outward bending motion between weak nail points. This process will consume energy. When the strain amplitude is large, dislocation damping is a static hysterical internal friction related to the strain amplitude. The entanglement of grain boundaries, second phases, and dislocations represent strong pinning points. When the critical strain amplitude is exceeded, dislocations will undergo an avalanche pinning motion at weak pinning points and continue to be pinned by strong pinning points. In magnesium alloys, alloy element atoms near dislocation lines will aggregate into gas clusters and exert pinning effects, resulting in a decrease in the density of movable dislocations in magnesium alloys. The effective dislocation length and the area crossed by the dislocation line during movement between strong and weak pinning points decrease.

In general, the damping characteristics of magnesium alloys are divided into strain amplitude-independent damping (Q_l_) and strain amplitude-dependent damping (Q_h_). Under low stress, the length between the dislocation segments is represented by L_c_. The following formula can be used to express the internal friction caused by overcoming resistance during movement:(1)$$Q_{l}^{-1}=\rho A\omega L_{c}^{4}/(36Gb^{2})$$

In the formula, ρ is the density of the movable dislocation, L_c_ is the length between the weak pinning points, G is the shear modulus, ω is the angular frequency, b is the Burger vector, and A is a constant. It can be seen that in the low strain region, the larger the length L_c_ between the dislocation density ρ and the weak pinning point, the better the damping performance of the alloy. The number of weak pinning should be appropriately reduced and higher dislocation density should be introduced.

Under high strain amplitude, the unraveling motion of dislocation consumes a lot of energy and produces large damping. The distance between dislocations varies from L_c_ (the distance between weak pinning points) to L_n_ (the distance between strong pinning points). At this time, the internal friction equation of strain amplitude can be written as:(2)$$Q_{h}^{-1}=(C_{1}/\varepsilon )\exp(C_{2}/\varepsilon )$$

In the formula, $C_{1}\sim \left(\frac{\rho L_{n}^{3}}{L_{c}^{2}}\right);C_{2}\sim (1/L_{c})$, ε is the strain amplitude and L_n_ is the distance between the strong pinning points. It can be seen that the damping performance of the alloy is inversely proportional to the length L_n_ between the strong pinning points, and increasing the strong pinning points will effectively improve the damping characteristics of the alloy [45].

According to the G-L mode theory, the damping energy consumption of magnesium alloy is worse than that of pure magnesium [46,47]. Alloying mainly regulates the damping performance of magnesium alloys through fine grain strengthening, solid solution strengthening, dislocation strengthening, and increasing the number of strong and weak pinning points.

Figure 3 shows the relationship between damping capacity and strain amplitude of several classic binary Mg-0.6 wt.% X magnesium alloys. The addition of different solid solution elements holds an impact on the damping capacity of magnesium alloys. Based on this, the mechanical properties of these magnesium alloys also differ. Therefore, studying damping magnesium alloys of various alloy systems and understanding their different properties play an important role in their applications.

### 2.1. Mg-Zr Damping Magnesium Alloys

This type of magnesium alloy is a traditional damping magnesium alloy mainly used in high-end fields such as aviation, diving, and modern weapons. Since the successful development of K1X1-F (with a Zr content of 0.6%) damping alloy and its improved version K1-A in the 1960s [49], it has attracted more and more attention due to its outstanding advantages such as excellent damping performance, excellent casting performance, fine grains, high liquid flow mode, and strong corrosion resistance [50,51,52,53]. Its damping performance is not only much higher than commercial aluminum alloys but also higher than gray cast iron. Its mechanical performance is also very excellent; σ_B_ reaches 175 MPa, and σ_0.2_ reaches 60 MPa. This can reduce the negative impact caused by vibration in precision aerospace instruments such as missile guidance, gyroscope compass, instrument chassis, etc. The Zr content holds a significant impact on the damping performance. As shown in Figure 4, when the Zr content is around 0.6 wt.%, the material exhibits the best damping performance under the same test conditions.

The addition of the Zr element has diverse effects on magnesium alloys. Grain refinement has been proven to be an important method for simultaneously improving the strength and ductility of metals and alloys [55,56]. The addition of the Zr element can greatly refine the grain size of magnesium alloys and improve their resistance to heat. Moreover, the content of Zr has a significant influence on the damping capacity and mechanical properties of Mg-Zr alloys. Rui-long Niu et al. [54] observed six ε-tan φ curves of Mg-Zr alloys with different Zr contents at room temperature and 373 K and found that there was a small increasing tendency in the damping capacity with an increase in stain ε under the condition of strain ε less than a certain value as the critical strain ε_cr_. Nevertheless, the internal friction tan φ, which expressed damping capacity, showed a greatly increasing trend with a rise in strain ε under the condition of strain ε outweighing the critical strain ε_cr_. Meanwhile, at the same temperature and strain amplitude, the damping capacity of Mg-xZr (x = 0.6, 1.0, 1.5, 2, 2.5, 5.0) alloys first increased and then decreased with the increase in Zr content. Among these alloys, Mg-0.6Zr exhibited the best performance, with the damping performance reaching an average of over 0.01 under experimental conditions. The damping mechanism of Mg-Zr alloys can be explained by the classical G-L theory.

To improve the mechanical property and damping capacity of Mg-Zr alloys, researchers have conducted a large number of experiments. Among them, heat treatment is one of the traditional methods. Ming-Hung Tsai et al. [57] conducted a series of temperature aging treatments on K1 alloy in the quenched state and found that when the temperature was 200 °C, the microstructure and damping capacity of as-quenched K1 alloys were not notably different. When the as-quenched K1 alloy was suffering from the aging treatment at 300 ℃, some twins were forming in the α-Mg grains. Within the scope of the experiment, those alloys that had aged at 300 °C for 16 h exhibited the best damping properties among all the samples in the experiment. The average ξ value reached 0.01724. When the aging treatment temperature reached 400 °C, coarse and loose twin structures were formed with the α-Mg grains. This would cause a decrease in the damping performance of the alloy. As the aging treatment temperature further increased to 500 °C, the microstructure of this alloy was comprised of both α-Mg and α-Zr phases, but the latter would not hurt the damping capacity of the K1 alloy.

The same classic approach is to add other elements to the alloy to enhance its properties. For example, Zn is a very important alloying element for Mg. The solid solubility of Zn in Mg is only 6.2% [58]. It is a very commonly used and practical magnesium alloy addition element. D.Q.Wan et al. [50] conducted tensile tests and studied the effect of Zn on the mechanical and damping properties of Mg-0.6% Zr. The result showed that the α-Mg grains could be refined by the addition of Zn, and the mechanical properties of the as-cast Mg-0.6%Zr-based alloys could be enhanced by the addition of Zinc. The tensile strength and the length increased with the rise in Zn content. This was the result of solid solution strengthening caused by the addition of the Zn element. However, the damping capacity of the Mg-0.6%Zr-0.5%Zn alloy decreased when the content of Zn decreased. The as-cast Mg-0.6%Zr-1.0%Zn exhibited the best mechanical properties among all the experimental samples—σ_b_ = 171 MPa and σ_0.2_ = 77 MPa. Its damping capacity was also excellent; its damping value was greater than 0.01, and it belonged to high-damping alloys. 

In summary, the research on the damping properties of Mg-Zr alloys shows that the damping properties of these alloys are better, and the addition of Zr can refine the grains and improve the strength of the alloy. A summary of the literature on these alloys is shown in Table 2.

### 2.2. Mg-Al Damping Magnesium Alloys

Al is one of the most pervasively used metal elements in Mg alloys and has a high solid solubility in Mg [59]. Adding Al as an alloying element to Mg is a very common method to increase the mechanical properties of Mg alloys. The Mg-Al alloy is currently the most widely used commercial metal, represented by the AZ91D alloy [60]. Its damping and shock absorption performance is good, and its shock absorption performance at room-temperature vibration frequencies is three times that of the 110OH18 aluminum alloy and nearly four times higher than that of quenched and tempered 60 steel. Its shock absorption performance increases with the increase in material grain boundaries and the second-phase Mg_17_Al_12_. Their applications are also very extensive, such as dashboard crossbeams, steering wheel armatures, and seat structures [27]. Currently, it is generally used for parts such as reduction cover materials for automotive engines, raw materials for motorcycle exhaust pipes, and materials for precision instruments such as instrument panels. After solution treatment, the damping properties of AZ31, AZ61, and AZ80 alloys decrease with the increase in Al content, depending on the length of aging time. Due to the aging-precipitated Mg_17_Al_12_ second phase and the interaction between the second phase and dislocations, alloys with high Al content have a better strengthening effects [61]. Its damping mechanism can be explained by the classical G-L theory.

Diqing Wan et al. [61] balanced the mechanical and damping properties of the Mg97Zn1Y2 alloy by adding a certain amount of Al element and found that the addition of aluminum had a significant refining effect on the grain size of the alloy. With the increase in aluminum content, a second phase of Mg_17_Al_12_ appears in the microstructure, which would hinder the growth of dendrites during the crystallization process. Meanwhile, with the increase in aluminum content, the mechanical properties of the alloy are improved. The addition of the Al element refined the grain size, and as its content increased, a second phase of Mg_17_Al_12_ was formed in the matrix. In this case, the grain boundary rapidly increased, hindering the movement of dislocations. Therefore, at low strain amplitudes, it was beneficial for damping performance. While at high strain amplitudes, excessive solute atoms will reduce damping performance. The cast structure, capacity, and mechanical properties of Mg-Al alloys were investigated by Mitsuharu Shimizu et al. [62]. The experiment presented a conclusion that the cast structure became finer as the content of Al increased, and the mechanical properties of the Mg-Al alloy improved a lot due to the addition of Al. However, the damping capacity of the Mg-Al alloy reduced when the content of Al increased further.

Adding other elements to Mg-Al alloys to regulate their mechanical and damping properties is also a very common method. Amit Kumar et al. [63] studied the effect of La element addition on the microstructure, mechanical properties, and damping performance of Mg-3Al alloy and found that the addition of La could refine grain size. As the La content increased, Al_2_La, Al_2.12_La_0.88,_ and Al_11_La_3_ intermetallic phases uniformly dispersed in the collective. Mg-3Al-2.5La showed the best strength and ductility. Its tensile yield strength reached 160 MPa, strain reached 22%, and the damping capacity of alloys increased with the addition of La. Hongbin Ma et al. [64] conducted an experiment of adding Y element to Mg-1Al-xY (x = 4, 6, 8) alloy to study the microstructure, mechanical properties, and damping properties of the alloy. The results showed that the grain size of the alloy decreased as the content of Y increased, and within the experimental range, the amount of Y element content does not change the morphology of the grains; they all present an equiaxed crystal. As the Y content increased, the mechanical properties of the alloy improved. Temperature has a significant impact on the damping performance of alloys. Between 300 K and 675 K, the overall trend of the damping performance of alloys decreases first and then increases with the increase in Y element content, and two damping peaks, P_1_ and P_2_, appeared at a certain temperature. They were caused by grain boundary relaxation and grain boundary slip, respectively. Joong-Hwan Jun [65] added Ca elements to the AZ91D casting alloy in an attempt to increase the damping performance of the alloy without affecting its mechanical properties. In the strain-amplitude-independent region, the size of Ca content did not have a significant impact on the damping performance of the alloy, while in the strain-amplitude-dependent region, the damping performance of the alloy decreased with the increase in Ca content. The AZ91-0.5%Ca alloy exhibited the maximum damping capacity. Joong-Hwan Jun [66] also compared the damping capacity of ZA82 and ZA64 alloys and the AZ91 alloy with the same mass fraction and found that the damping capacity of ZA alloy was higher than that of AZ alloy in both strain-amplitude-dependent and strain-amplitude-independent regions. Moreover, by comparing the two alloys of ZA in Figure 5, they found that the ZA82 alloy had superior damping capacity compared to the ZA64 alloy.

It is also a regular method to regulate the damping capacity of alloys by influencing factors such as the structure, quantity, and morphology of internal defects in the alloy. The experiment of conducting small tensile deformation on magnesium alloys to affect damping performance was conducted by Hu Xiao-shi et al. [67]. The casting Mg-1%Al alloy possessed a high damping capacity at room temperature. In the tensile experiment, especially when the tensile elongation was greater than 3%, the damping capacity of Mg-1%Al in the high-strain region decreased significantly. This was because of the decrease in length and movability of the dislocations at room temperature. When the temperature reached a certain level, plenty of dislocations caused by tensile deformation were activated and able to move in a short time, and energy was consumed during this process. It led to an increase in damping value.

Mg-Al series alloys are one of the most widely used alloys in all magnesium alloys. They have good mechanical properties and high damping properties. Summary of the researches of damping properties and mechanical properties of this kind of alloy is shown in Table 3

### 2.3. Mg-Ni Damping Magnesium Alloys

As a high-damping material, the Mg-Ni damping magnesium alloy is a hypoeutectic Mg-Ni alloy, whose microstructure is composed of α-Mg solid solution and eutectic phase (α-Mg + Mg_2_Ni) [68]. As early as the 1970s, Japanese scientist Sugimoto [32] found that adding a small amount of metal Ni to pure magnesium could significantly improve its mechanical properties, and when the addition amount was about 0.1%, the damping performance of the alloy did not decrease but increased instead [69]. Recently, there has been new progress in research on Mg-Ni-based damping magnesium alloys. The damping capacity of the Mg-Ni damping magnesium alloy is relatively good; for instance, at an amplitude of 1 × 10^−3^, the specific damping coefficients of Mg-3%Ni, Mg-13%Ni, and Mg-23.5%Ni are 40%, 26%, and 4%, respectively. But usually, the damping performance and corrosion resistance of this series of alloys are difficult to balance, and ensuring damping performance often requires sacrificing its corrosion resistance [70]. By adding different contents of Mn to Mg-3Ni alloys, Wan et al. [71] found that the corrosion resistance and the damping capacity after exceeding critical strain increased as the content of Mn increased. This could provide a good method for improving the damping performance of these alloys in the future.

In recent years, research on Mg-Ni-based damping magnesium alloys has been more detailed. Hu X S et al. [72] found that there were two damping peaks, P_1_ and P_2_, in both the Mg-Ni alloy and pure Mg at 100 °C and 230 °C, where P_1_ was caused by dislocation motion. As the temperature increased, the rearrangement of impurity atoms inside the alloy caused a decrease in P_1_, while the P_2_ was caused by grain boundary slip. Wan et al. [73] found that the hypoeutectic Mg-Ni alloys exhibited high strain amplitude-dependent damping properties. Several damping plateaus were found in the damping curve for Mg-3%Ni and Mg-6Ni alloys. When the content of Ni increased, the tensile strength and elastic modulus increased, but the ductility decreased, which was because of the increase in brittle Mg_2_Ni. At room temperature, there was no energy dissipation at the interface of the cast hypoeutectic Mg-Ni alloy. The decrease in damping value of high Ni alloy was due to the decrease in volume fraction and size of α-Mg dendrites. R. Schaller et al. [74] studied the damping and mechanical properties of directional solidification Mg-Ni alloys. They found that with the increase in Ni content, the mechanical properties of the alloy increased, but the damping properties declined, and the degree of damping decline was less than the degree of mechanical property improvement. Moreover, the mechanical properties of the Mg-10% Ni alloy and the Mg-15% Ni alloy were comparable to AZ63 (Mg-6%Al-3%Zn), but the damping performance was 30 times that of AZ63.

Improving the structure and properties of Mg-Ni alloys through heat treatment is a recent focus of research on Mg-Ni alloys [75]. Heat treatment can readjust the density of dislocations or other defects in the material, eliminate disordered areas, and improve the damping performance of the alloy [76,77]. Ruopeng Lu et al. [78] studied the microstructure, mechanical properties, and damping properties of Mg_95.34_Ni_2_Y_2.66_ alloys and Mg_95.34_Zn_1_Ni_1_Y_2.66_ alloys, especially the difference in the long-period stacked ordered phase (LPSO) in these two alloys during heat treatment. After two hours of heat treatment at 773 K, the eutectic phase was integrated into the matrix, and the LPSO phase still maintained a constant structure. Dislocation internal friction peaks and grain boundary internal friction peaks were found after temperature-dependent damping of the Mg_95.34_Ni_2_Y_2.66_ and Mg_95.34_Zn_1_Ni_1_Y_2.66_ alloys. After heat treatment, damping peaks were found to increase in both of the two samples, especially in Mg_95.34_Ni_2_Y_2.66_. The annealed Mg_95.34_Ni_2_Y_2.66_ alloys with a rod-shaped LSPO phase showed a good damping capacity and exceeded the high damping standard (Q^−1^ > 0.01). This might be caused by the difference between the second phase and solid solution atom content.

As shown in Table 4, the damping properties and mechanical properties of Mg-Ni alloys are summarized. From the current literature, although the damping performance of Mg-Ni alloy is good, the corrosion resistance of Mg-Ni alloy is poor, and it still needs further study as an engineering damping material.

### 2.4. Mg-Li Damping Magnesium Alloys

The research on Mg-Li-based alloys began in 1910 when German scientists Masing et al. [79] discovered that Li could alter the crystal structure of Mg. When the Li content exceeded 5.7%, the crystal structure of magnesium metal would transition from a hexagonal close-packed (HCP) structure to a body-centered cubic (BCC) structure [80]. Research [81] on the Mg-Li binary system showed that when the content of Li was higher than 5.7%, Li atoms would exist in the alloy in the form of solid solution atoms and form a single solid solution α-Mg; when the content of Li was higher than 10.3%, Mg atoms would dissolve in Li and form a single solid solution β-Li; and when the Li content was between 5.7% and 10.3%, the alloy comprised of α- Mg and β-Li’s bi-directional organization [82]. As a phase metal element, Li can not only greatly improve its formability but also greatly reduce its density [82,83,84]. That is why the Mg-Li alloy is called an ultra-light alloy. Meanwhile, there are literature reports [82,85] reporting that the addition of Li can improve the elastic modulus of magnesium alloys with low lithium content. This is because of the relationship between α-Mg matrix and the solid phase solution of Li in the Mg phase matrix. In addition, the addition of Li can activate non-substrate dislocations, thereby improving damping capacity. Wang et al. [86] found that the addition of the Li element could change the crystal structure of the alloys, thereby affecting the damping capacity of the alloy and founding a new type of damping mechanism. Suyash Kumar Mishra et al. [14] made Mg-8.4Li alloy and Mg-9Li alloy through disintegration melting deposition. They found that the two alloys were superior to the existing commercial magnesium alloys in terms of specific strength, strain hardening ability, and ductility. The fracture strain of the two alloys was greater than 80%. Among them, the Mg-8.4Li alloy had the highest compressive yield strength of 192 MPa, and the Mg-9Li alloy showed the highest ultimate compressive strength of 2312 MPa. Compared with pure magnesium, the damping capacity of the Mg-8.4Li alloy increased by 62%, while the damping capacity of the Mg-9Li alloy increased by 18%. Compared with pure magnesium, the attenuation loss rate of Mg-8.4Li and Mg-9Li increased by 275% and 141%, respectively. Compared with pure magnesium, the elastic modulus of Mg-8.4Li and Mg-9Li also showed an increase of 11% and 13%. The results showed that the addition of lithium increased the elastic modulus of the binary alloy but reduced the damping capacity. Yet the damping capacity was still better than that of pure magnesium. However, some defects also emerged when adding Li into Mg alloys. First of all, the addition of Li will reduce the strength of the alloy [87]. As the content of Li exceeds a certain level, the elastic modulus will bring a negative effect on the mechanistic capacities of the alloys because Li’s elastic modulus is only 8 Gpa. Although the Mg-Li alloy has good plasticity, the mechanical properties of this binary alloy are always low, with a dual structure alloy (α-Mg + β-Li) that has a tensile strength of 110–120 MPa and a yield strength of 60–90 MPa at room temperature, the tensile strength of the single-phase structured (β-Li) Mg-Li alloy is around 100 MPa, and the yield strength is even lower than 60 MPa. In addition, the high-temperature mechanical properties of binary alloys are also poor and can lead to creep failure under small stresses [81,88], and they are very difficult to utilize alone.

Therefore, other metal elements are generally added for further alloying to form ternary or multicomponent alloys to regulate mechanical properties [89]. Currently, research on the damping properties of Mg-Li alloys is mostly focused on adding other metal elements. For example, Al is a typical alloying element in Mg-Li alloys, and it can increase the strength and hardness of Mg-Li alloys without sacrificing their density and damping properties [90]. A total of 4% Al was added to Mg-5Li by N Rahulan et al. [83], and the as-extruded Mg-5Li-4Al alloy exhibited a high specific strength of 145 kNm/kg. On this basis, Gang Zhou et al. [48] prepared a high-strength and high-damping Mg-4Li-3Al-0.3Mn alloy by adding trace amounts of Mn to the Mg-Li-Al alloy and combining it with a hot extrusion. The as-extruded alloys exhibited good mechanical properties. Its yield strength, tensile strength, and elongation were 248 MPa, 332 MPa, and 14.3%, respectively. Meanwhile, the specific strength of the alloy was 202 kNm/kg. Grain boundary strengthening, dislocation strengthening, and precipitation strengthening were the principal strengthening mechanisms.

After extruded, the damping capacity of Mg-4Li-3Al-0.3Mn alloys increased from 0.022 to 0.028 at a strain amplitude of 1 × 10^−3^ and showed a good damping performance. The damping mechanism at room temperature can be explained by the G-L theory. After the temperature rose to a certain extent, in addition to dislocation damping, phase and grain boundary damping also contributed to the damping. Song et al. [91] studied the mechanical and damping properties of Mg-Li-Zn alloys, and the results showed that the strength of the alloy increased with the increase in Zn content, while the plasticity decreased with the increase in Zn content. At the same time, the addition of Zn can improve the ductility and damping capacity of Mg-Li alloys.

Jing-feng Wang et al. [86] found that by regulating the content of two coexisting binary Mg-Li alloys and studying their mechanical and damping properties. They found that no matter how the content of the Li element changed within the experimental range, a second phase would not be formed. Mg and Li atoms in the alloy existed in the form of solid solution atoms. In the strain-independent stage, the content of the Li element has little effect on damping, while in the strain-related stage, damping capacity was poorer with a greater amount of β-Li phase. At the same time, due to the lack of a slip system in the hexagonal close-packed structure of α-Mg, damping capacity would become impeded. Therefore, when there was no α-Mg phase in the alloys, the alloys would exhibit an extremely high damping capacity. Moreover, the G-L curve of the Mg-xLi binary alloy with two coexisting terms was not a straight line, which indicates that its damping mechanism was not simply a G-L dislocation mechanism. It was a new damping mechanism generated by the combined effect of dislocation damping at room temperature and dislocation damping at high temperatures with phase boundary damping. The contribution of phase boundary damping to macroscopic damping was also quite significant. A Mg-Li alloy coexisting in two phases has more stable physical, mechanical, and damping properties than a single-phase alloy, making it a high-damping material that is both lightweight and has shock absorption and noise resistance. Yang Xinhe et al. [92] found through research on the effect of solid solution hot rolling aging on the mechanical properties of the Mg-13Li-3Al-3Zn alloy and Mg-13Li-3Al-6Zn alloy that although solid solution strengthening significantly improved the hardness of the alloy, it greatly reduced ductility, and the Al and Li softening phases precipitated during subsequent hot rolling and failure processes could greatly enhance the alloy’s recovery. However, excessive precipitation of the Al-Li softening phase and solid solution of excessive Zn element was not conducive to the mechanical and damping properties of the alloy. Subsequent aging treatment could reduce the number of solute atoms and significantly improve mechanical and damping properties. The tensile strength and elongation of Mg-13Li-3Al-3Zn obtained from the results were 184 MPa and 32%, respectively. The aging state had higher damping than the casting state. In the low-strain amplitude stage, the damping value was greater than 0.014, indicating that it belongs to a high-damping material. The improvement in damping ability was mainly due to the decrease in the number of solute atoms in the matrix. This led to an increase in the distance of dislocation movement. Wang Dan et al. [93] prepared Mg-8Li-4Y-2Er-2Zn-0.6Zr alloy with high damping, high elastic modulus, and high mechanical properties by using heat treatment and cold rolling methods. After heat treatment at 18 °C for 450 h, the LPSO phase was introduced into the alloy. After cold rolling, the LPSO phase underwent twinning and twisting phenomena, increasing the damping capacity from 0.01 to 0.02 and increasing the tensile strength from 142 MPa to 221 MPa. The elastic modulus of the cold-rolled alloy reached 48.9 GPa, making it a material with excellent comprehensive properties. Conducted by Jiahao Wang et al. [94], a Mg-8Li-6Y-2Zn alloy with high damping capacity and sufficient strength was prepared by using heat treatment and hot rolling methods. The mechanical properties and damping capacity of the Mg-8Li-6Y-2Zn alloy under different states were systematically studied. Compared with casting alloys, the tensile strength of the alloy treated with heat treatment and rolling increased from 140 MPa to 210 MPa, and the damping capacity increased from 0.006 to 0.018.

As shown in Table 5, the research on the mechanical properties and damping properties of Mg-Li alloys is summarized. Mg-Li alloys have become a research hotspot of damping magnesium alloys in recent years due to their superior light weight and machinability. However, the mechanical properties of this series of alloys are poor, and further development is needed for engineering materials.

### 2.5. Mg-RE Damping Magnesium Alloys

Table 6 shows the solid solubility of some RE elements in Mg and the compounds that can be formed with Mg. Adding rare earth elements to Mg alloy can not only significantly improve the mechanical properties of the alloy at room temperature and high temperatures but also improve the casting properties of the alloy and refine its structure [37]. Its special electronic structure and obvious strengthening effect have made Mg-RE alloys become the focus of research in recent years [95].

Y, with a relatively large solid solubility in the Mg matrix, is a typical adding element. The addition of the Y element can significantly refine the grains and improve the corrosion resistance of alloys [96]. At the same time, the addition of the Y element can also improve the high-temperature tensile and compressive properties and creep properties of the alloy [97]. By adding a small amount of Y element, the overall damping capacity of Mg and the overall static/dynamic/ignition response of Mg can be improved [98]. Y. T. Tang et al. [99] studied the damping capacity and mechanical properties of extruded sheet Mg-xY (x = 0.5, 1.0, 3.0 wt.%) in detail. They found that the damping capacity of the alloy decreased from 0.037 to 0.015 with the content of Y element rising from 0.5% to 3.0% at room temperature, and the damping mechanism was found to be consistent with the G-L dislocation theory. With the increase in temperature, the G-L curve gradually deviated from linearity, which meant that the damping mechanism of Mg-Y alloy at high temperatures not only obeyed the G-L theory but also the grain boundary damping mechanism. This result showed that the segregation of the Y element at the grain boundary could inhibit grain boundary sliding. Among them, the extruded Mg-1Y alloy sheet exhibited higher yield strength and damping properties. At 325 °C, the damping capacity of the alloy was close to that of pure magnesium, but the yield strength was more than three times that of pure magnesium. Rui-long Niu et al. [100] studied the Mg-0.6Zr alloy’s microstructure after adding the Y element to it. They found that with the increase in Y element content (mass fraction from 1.0% to 5.0%), the grain size of the α-Mg matrix decreased obviously. There was a Mg_24_Y_5_ phase rich in the Y element. They found that when ε > ε_cr_ = 2.0 × 10^−2^, the damping of the alloy would be greatly affected, but when ε was less than ε_cr_, the damping capacity would not be greatly affected. The damping mechanism obeyed the G-L model. At the same time, when the temperature rose from ambient temperature to 673 K, the damping of the alloy also showed an upward trend; this could be well explained by the grain boundary damping mechanism. In addition, as the content of the Y element increased (mass fraction from 1.0% to 4.0%), the yield strength and ultimate tensile strength of the alloy increased, but when the content of the Y element was further increased (mass fraction from 4.0% to 5.0%), the mechanical properties of the alloy would not be further improved. Among them, the Mg-0.6Zr-4.0Y alloy had the best damping capacity and mechanical properties. In recent years, there have been many studies on Mg-Y series damping magnesium alloys. Among them, heat treatment and introducing the LSPO phase were found to be two very effective ways to regulate the alloys’ capacities. Dan Wang et al. [101] studied the transformation of the LSPO phase during heat treatment of the Mg-4Y-2Er-2Zn-0.6Zr alloy and its effect on damping and mechanical properties. They found that the alloy was initially composed of α-Mg and lamellar 14H LSPO phase. After heat treatment at 510 °C for 8 h, these lamellar 14H LSPO phases gradually transformed into a massive 18R LSPO phase that could improve the damping capacity and mechanical properties, whether at a low or high temperature. Dan Wang et al. [102] improved the comprehensive mechanical properties and damping capacity of the Mg-Y-Er-Zn-Zr alloy by extrusion, heat treatment, and rolling. The tensile strength of the rolled alloy reached 362 ± 5 MPa, and the elongation reached 7.8 ± 0.4%. After heat treatment, the group of the alloy has also been greatly improved. In the matrix and LSPO phase, the generation of twins improved the damping of the rolled alloy and eliminated the negative effect of strong pinning points on damping. When the strain strength was ε = 10^−1^, the damping of the rolled alloy reached 0.018. By changing the content of the Y element in Mg-xY (x = 1, 3, 7 wt.%) alloys, as well as T4 heat treatment and pre-deformation of the material, L. B. Ren et al. [75] found that when the content of the Y element increased from 1 wt.% to 7 wt.% in the strain range of 10^−6^~10^−3^, the damping performance of the material decreased significantly from 0.15 to 0.002, while the yield strength increased from 20 ± 2 MPa to 92 ± 4 MPa, and the damping mechanism of the alloy conformed to the G-L model. After T4 heat treatment, the damping performance of the alloy was significantly improved, which was speculated that more pinned Y atoms were distributed in dislocations. The pre-deformation of 25% reduced the damping properties of Mg-1Y alloy after T4 heat treatment. This was because of the increase in dislocation density caused by work hardening.

Mg-Sc alloys have been reported to have very good shape memory properties [103,104]. When the content of Sc exceeds 18 at%, metastable body-centered cubic single crystals can be obtained [105]. The Mg-SC alloy with two α + β phases exhibits a good balance between tensile strength and elongation, achieving high tensile strength greater than 300 MPa and excellent elongation of about 30%. Among these, the α phase is a densely packed hexagonal phase (hcp) in Mg alloy, and the β phase is the body-centered cubic phase (bcc) in magnesium alloys [106]. Ogawa et al. [107] studied the damping capacity of the hexagonal close-packed phase, body-centered cubic phase, and martensite phase in Mg-Sc alloy by nano-dynamic mechanical analysis equipment. They found that the damping capacity of the alloy in the martensitic period was significant. The lath martensite contained some lattice defects, so the damping capacity is higher than that in the body-centered cubic period, and the close-packed hexagonal period of martensite was considered to be caused by the movement of lattice defects such as double interfaces.

The Gd element is considered one of the most effective rare earth elements to improve the mechanical properties of magnesium alloys [108]. Its strengthening effect comes from solid solution strengthening because the atomic radius of the Gd element differs significantly from that of the Mg element, and the solid solution limit range is wide [109], so Gd is considered an even more effective strengthening element than the universally acknowledged elements such as aluminum, zinc, and manganese [110,111]. Yajie Ma et al. [112] prepared a new Mg-1.5Gd binary damping alloy with ultra-high ductility and sufficient strength by hot extrusion at different temperatures of 360 °C, 420 °C and 480 °C and studied its microstructure, damping properties, and mechanical properties. They found that the tensile elongation of the alloys exceeded 40%, and the damping value reached 0.086 at ε = 10^−3^. Under the same test conditions, the damping value and yield strength of the Mg-1.5Gd alloy extruded at 360 °C were higher than those of the Mg-1Y alloy. Chen Su et al. [113] studied the microstructure, damping properties, and mechanical properties of Mg-8.5Gd-5Y-xAl (x = 0.2, 0.5, 0.8, 1.1 wt.%) alloys. They found that the alloy was mainly composed of the LSPO phase, Mg-RE phase, Al-RE phase, and α-Mg matrix. With the increase in Al content, the LSPO phase gradually decreased. After extrusion, the Mg-8.5Gd-5Y-0.2Al alloy had the best mechanical properties and damping properties. The tensile strength was 376 MPa, the yield strength was 263 MPa, and the elongation was 13%. When the strain was 10^−3^, the Q^−1^ was 0.0132. Yajie Ma et al. [114] found that the addition of Gd to Mg could form a parallel distribution of the second phase after solidification, resulting in a large number of parallel dislocations in the α-Mg matrix of Mg-Gd. This could effectively slow down the decrease in damping performance caused by the increase in element content. When the Gd content increased from 1 wt.% to 6 wt.%, the Q^−1^ decreased from 0.16 to 0.05, and the yield strength increased from 30 MPa to 84 MPa when the strain amplitude was 0.8 × 10^−3^. Considering the balance between damping and mechanical properties, the Mg-Gd alloy could be used as a potential candidate material for the development of damping magnesium alloys.

The solubility of Ce in Mg is relatively low and will not affect the corrosion resistance of the alloy. Ce can increase the performance of magnesium alloys at room temperature [115]. At the same time, the addition of Ce can greatly refine the grain, and the yield strength of magnesium alloy is greatly improved [116]. In addition, Ce can produce a dendritic Mg_12_Ce phase with high thermal stability in magnesium alloy. Therefore, Ce as the additional element in a magnesium alloy is very meaningful. Zhongshan Wu et al. [117] found that many parallel second phases in Mg-2Ce alloys show excellent damping capacity in the strain amplitude and temperature-dependent region. At ε = 10^−3^ and room temperature, the damping value is 0.018. In Mg-0.5Ce and Mg-1Ce, two damping peaks, P_1_ and P_2_, were found at 78 °C and 167 °C, respectively, but only one dislocation peak, P_1_, appeared in the Mg-2Ce alloy. In the Mg-2Ce alloy, there are many dislocations around the special parallel Mg_12_Ce phase, but in the Mg-1Ce alloy, there were only a few dislocations around the network Mg_12_Ce phase. In the α-Mg matrix of the Mg-2Ce alloy, the rich and uniform parallel dislocation configuration played an important role in the improvement of damping. A low-cost Mg-1Al-0.5Ce alloy with comprehensive properties was successfully prepared by casting, heat treatment, and hot extrusion [118]. Its tensile yield strength reached 169 MPa, and Q^−1^ reached 0.035.

For Mg-RE alloys, the properties of this series of alloys are better. For different rare earth elements, different second phases will be formed in magnesium alloys, and most rare earth elements can enhance the mechanical properties of magnesium alloys, among which Y, Gd and Er are the most prominent. As shown in Table 7, the mechanical properties and damping properties of Mg-RE series alloys are summarized.

### 2.6. Mg-Si Damping Magnesium Alloys

The Mg-Si damping magnesium alloy is a damping alloy prepared by directional solidification around the 1990s. It is found that Mg-1.34%Si and Mg-2%Si alloys have good damping properties and mechanical properties. It is speculated that the solubility of Si in Mg is very small and will not corrode Mg like Ni elements [119]. So, it does not affect the damping properties of the alloy, and the formed strengthening phase can greatly improve the mechanical properties of the alloy. Recent studies [120,121] reported that the maximum solid solubility of Si in Mg is only 0.003 at %, and Si atoms could react with Mg atoms to form a Mg_2_Si intermetallic compound, which was a stoichiometric stable material with a high melting point (1085 °C), low density (1.99 × 10^3^ Kg/m^3^), high hardness (4.5 × 10^9^ N/m^2^), low expansion coefficient (7.5 × 10^−6^ K^−1^), and high elastic modulus (120 GPa). The mechanical properties, heat resistance, and wear resistance of Mg-Si alloys have been greatly improved. However, the size of Mg_2_Si particles is too large, resulting in lower strength and ductility, as well as higher brittleness [122]. So, the performance of magnesium silicon alloys is generally improved by reducing the particle size and then refining the microstructure [123].

Figure 6 shows the optical microstructures of as-cast Mg-Si alloys. Figure 6a,b are the micrographs of hypoeutectic Mg-0.3 wt.%Si and Mg-0.8 wt.%Si alloys, respectively. They consist of large dendrite or globular primary α-Mg and eutectic phase, which is fine needle-shaped Mg-Mg_2_Si, as shown in Figure 6c. For these hypoeutectic Mg-Si alloys, the increasing Si content reduces the size of the primary α-Mg but markedly increases the amount of the eutectic Mg-Mg_2_Si phase. The average size of primary α-Mg in Mg-0.3 wt.%Si and Mg-0.8 wt.%Si alloys is about 200 and 120 μm, respectively, and the space of Mg between the two needle-shaped Mg2Si is 1 μm. For the hypereutectic Mg-2.3 wt.%Si alloys, the in situ formed polygonal primary Mg_2_Si particles with an average dimension of 15 μm and needle-shaped eutectic phase Mg-Mg_2_Si are shown in Figure 6d,e. The space of Mg between two needle-shaped Mg_2_Si in this eutectic phase is about 4 μm, which is much larger than that in hypoeutectic Mg-Si alloys. Besides that, there are still large amounts of Mg areas (white areas) that are free of the Mg_2_Si phase, and the mean size of these Mg areas is about 70 μm. All the polygonal primary Mg_2_Si particles are surrounded by Mg halos with dimensions of 10–30 μm.

The damping performance of the Mg-Si damping magnesium alloy is relatively high and has a great relationship with the content of Si. X. S. Hu et al. [124] studied the effect of Si content on the damping capacity and mechanical properties of cast Mg-Si alloys. They found that when the Si content was 0.3 wt.%, 0.8 wt.%, and 2.3 wt.%, the damping capacity of the alloy was good, and the damping value Q^−1^ was greater than 0.01. This was because these magnesium alloys possessed a large proportion of the magnesium phase and less soluble atoms. However, the tensile strength of Mg-0.8 wt.%Si alloys was greater than that of Mg-2.3 wt.%Si alloys because there were more brittle Mg_2_Si particles in the latter, resulting in stress concentration and earlier failure. The damping properties of Mg-Si alloys, whether hypoeutectic or hypereutectic, were good. However, the former was higher than the latter. It is speculated that the reason may be that the too-coarse Chinese character-like Mg_2_Si phase in the alloy easily splits the brittleness of the matrix and affects the damping performance. In the Mg-0.3wt.%Si alloy, the content of the Mg_2_Si phase was the least, and the content of the α-Mg phase was the most. In this alloy, the dislocation density was the smallest, and the dislocation length was relatively long. This was the reason why the damping capacity in the region with small strain was the minimum, and the damping capacity in the region with high strain was the maximum. They also found that two damping peaks, P_1_ and P_2_ [124,125], were detected in the temperature-dependent damping of the Mg-Si alloy and pure magnesium. The reason for P_1_ [124] was considered to be the interaction between dislocations and point defects, and P_2_ [124,125] was caused by grain boundary slip.

However, the high damping capacity of the hypoeutectic Mg-Si alloy is not stable. X. S. Hu et al. [119] found that the damping capacity of hypoeutectic Mg-Si alloys was very sensitive to heat treatment temperature and time, which would influence point defects. No matter what heat treatment conditions were used, a small amount of soluble impurity atoms would provide damping and ensure that the damping was in a relatively high position. And heat treatment also had an effect on the two damping peaks mentioned above. When the alloy was heat treated, the distribution of point defects would change and affect the height of these damping peaks.

### 2.7. Mg-Cu-Mn Damping Magnesium Alloys

Previous studies have shown that Mn can reduce the solubility of Fe in alloys, thereby improving the overall corrosion resistance of the alloy and refining the grain size of Mg [126,127,128]. Cu can improve the room-temperature and high-temperature strength of alloys, but it will reduce their corrosion resistance and ductility [129]. Therefore, Cu is generally added to alloys such as Mg-Mn-Zn [130]. The Mg-Cu-Mn alloy system was developed at the end of the twentieth century. Nishiyama et al. first discovered that the as-cast Mg- (0.5~0.7%)Cu-(0.17~4.0%)Mn alloys possessed good casting properties, corrosion resistance, and cutting performance. On this basis, they began researching a new Mg-Cu-Mn damping alloy. In 2003, Nishiyama et al. [10] prepared a CM31 alloy by powder metallurgy. They found that the addition of Mn improved the damping performance of the alloy. The specific damping value reached 60% and was much higher than that of the AZ91 magnesium alloy. When the strain amplitude was greater than 4 × 10^−5^, the damping capacity was even higher than that of pure magnesium, and tensile strength reached 290 MPa. On this basis, Zhenyan Zhang et al. [131] studied the effect of Cu and Mn additions on the mechanical properties and damping properties of Mg-Cu-Mn alloys. They found that the addition of Cu and Mn could significantly reduce the grain size, and grain refinement was the main reason for changing the mechanical properties. They also found that the Mg-2.5Cu-0.8Mn alloy possessed the best comprehensive properties. Its yield strength was 43 MPa, ultimate tensile strength was 131 MPa, elongation was 6.81%, and the Q^−1^ was more than 0.01 at ε = 10^−3^. By adding Cu to the Mg-1%Mn alloy and studying its microstructure, mechanical properties, and damping properties, Shuqun Chen et al. [132] found that the addition of Cu could significantly reduce the grain size of Mg-1%Mn and form spherical Mg_2_Cu in the matrix grains. The precipitation of a thick layered Mg_2_Cu phase at the grain boundary would deteriorate the mechanical properties of the alloy and promote the possibility of grain boundary fracture of the alloy. In the case of low strain amplitude, the increase in Cu content almost did not affect the damping capacity of the alloy. At high strain amplitude, the damping capacity of the alloy decreased with the decrease in grain size and the increase in the precipitated phase.

The performance of the alloy can be further improved by processing the alloy. Conducted by Mingyi Zheng et al. [133], the Mg-3%Cu-1%Mn alloy was subjected to four equal channel angular pressing (ECAP) treatments at 250 °C. They found that the grain size of the alloy was refined, the yield strength and tensile strength decreased, but the ductility was improved, and the damping capacity of the alloy was reduced at room temperature. After annealing at 300 °C for 1 h, the yield strength and tensile strength of the alloy further decreased, but the plasticity was significantly improved, and the damping performance was also restored. In addition, at a higher strain amplitude, the damping performance was better than that of the extruded alloy. After equal channel angular pressing treatment, the grain size of the alloy was small, and the grain boundary was unbalanced. When the temperature was high, the alloy showed better damping capacity. However, after annealing, the damping capacity decreased due to grain growth and equilibrium grain boundaries.

WANG Jing-feng et al. [134] studied the effects of Y and Zn on the mechanical properties and damping properties of Mg-Cu-Mn alloys. The addition of Y and Zn elements could reduce the grain size of the alloy and increase the yield strength of the alloy. With the increase in Y and Zn contents, the metal compounds would precipitate along the grain boundary and result in a decrease in the plasticity of the alloy. The damping property Q^−1^ of the alloy was greater than 0.01, and the alloy belonged to the high-damping alloy category. When the strain amplitude was 6 × 10^−5^, the damping performance of Mg-3Cu-1Mn alloy reached 0.01, and the damping performance was very good. When the strain amplitude was greater than 5 × 10^−4^, the damping capacity of the Mg-3Cu-1Mn-1Y-2Zn alloy increased the most due to the increase in the density of movable dislocations. The damping capacity of the Mg-3Cu-1Mn-1Y-2Zn alloy was even close to pure Mg. However, the tensile strength and yield strength of the alloy were not high, being only 71 MPa and 47 MPa, respectively. This was because as the content of Y and Zn elements increased, brittle compounds precipitated at the grain boundaries, resulting in brittle fracture during the tensile process. Meanwhile, Jingfeng Wang et al. [135] added a long-period stacking ordered (LSPO) structure to the Mg-Cu-Mn-Zn-Y alloy and studied its microstructure and damping capacity. The results showed that the LPSO structure was generally beneficial to the yield strength and damping capacity of the alloy. When the strain was greater than 1.1 × 10^4^, the damping Q^−1^ of the alloy was greater than 0.01. The alloy could be classified as a high-damping alloy. When many LSPO phases were observed in the alloy, the simultaneous increase in damping capacity and yield strength broke the contradiction between mechanical properties and damping properties of magnesium alloys. In addition, the damping mechanism of the alloy with the LSPO phase could not be only explained by the G-L dislocation theory. It is reported that they discovered a new frequency-dependent damping mechanism, temporarily named “LSPO mechanism”.

As shown in Table 8, the damping properties and mechanical properties of Mg-Cu-Mn alloys are summarized. The addition of Cu and Mn to Mg can produce different effects, which have a good inhibitory effect on the original harmful elements in Mg alloy and improve the performance. The mechanical properties and damping properties of these alloys are good, and the damping mechanism is not only the dislocation damping mechanism, but also has great research value.

## 3. High-Damping Magnesium-Based Composite Material

Since Mg is an active metal, the research on Mg composites requires a more stringent processing environment. Therefore, in the early days of research, the damping properties of magnesium metals were often developed in the direction of alloying. It has been found that adding reinforcements to pure magnesium or magnesium alloys can improve the damping properties of metals. This is because the dislocations in the magnesium matrix composites include not only the dislocations in the matrix but also the dislocations in the reinforcement [136]. At the same time, due to the mismatch of the thermal expansion coefficient between the matrix and the reinforcement, the damping performance of the magnesium-based metal material is greatly affected by the temperature. When the temperature changes, the thermal mismatch residual stress will be generated near the interface and result in the plastic flow of the whole matrix. Then, the dislocation density will be further increased and result in an increase in damping performance. From the 1950s to the 1960s, people began to conduct a lot of research on metal matrix composites. However, because Mg was too active, it was difficult to overcome this difficulty at that time. Most of the research focused on copper-based and iron-based composites. Crawley et al. [137] prepared graphite fiber-reinforced magnesium matrix composites in 1986. They first studied the damping properties of the materials. The experimental results showed that the damping properties were good. In 1986, Goddard et al. [138] studied the damping properties of non-directional graphite fiber-reinforced magnesium matrix composites. They considered that the damping properties of composites were similar to the Zener curve of the magnesium alloy AZ91C, but the damping of composites was higher than that of AZ91C alloys. In the 1990s, with the development of composite material preparation technology and the increasing demand for high-damping, high-strength, and low-density damping materials, the research of magnesium matrix composites had become more and more popular, and the research on its damping performance was also increasing.

The damping properties of magnesium matrix composites are usually better than those of unreinforced alloys. This can be demonstrated in the experiment conducted by Sourav Ganguly et al. [139]. Li-Hua Liao et al. [140] improved the damping properties of Mg_2_Si/Mg-9Al composites by using a hard metal compound, Mg_2_Si. They found that Mg_2_Si/Mg-9Al composites had better damping properties than non-reinforced alloys. The damping properties of the composites were further improved after the morphology modification of Mg_2_Si particles. Wei Cao et al. [141] studied the damping properties of in situ TiC-reinforced AZ91D composites with different reinforcement ratios. They found that TiC-reinforced magnesium matrix composites possessed better damping properties than unreinforced magnesium alloys. The damping properties of the materials increase with the increase in the volume fraction of TiC particles. This could be attributed to the dislocation damping mechanism of the composite at room temperature and the common mechanism of interface damping and dislocation damping at high temperatures. Xiuqing Zhang et al. [142] prepared TiC-particle-reinforced AZ91 magnesium matrix composites using an in situ synthesis method and studied the mechanical properties and damping properties of the composites. They found that the addition of TiC particles could refine the grains and increase the dislocation density. This had an important influence on the damping and mechanical properties of the composites. Compared with the AZ91 alloy, the damping capacity and tensile strength of the composites were improved. Wenbo Yu et al. [143] studied the damping properties of AZ91D composites reinforced by tensile Ti_2_AlC particles. They found that as the volume fraction of Ti_2_AlC particles increased from 0% to 20%, the damping capacity increased. This was because Ti_2_AlC itself had a large dislocation density and its damping conditions were good. This was beneficial to the overall damping capacity of the alloy. With the increase in the Ti2AlC particle volume fraction, the increase in interface damping would increase Q^−1^ significantly under high strain conditions. Above 200 °C, the damping properties of the composites increased obviously due to the interface sliding. At the same time, compared with magnesium alloy and pure magnesium, magnesium-based composite foam material had better damping capacity.

The interaction between the enhanced phase and matrix interface also has an impact on damping. Yu et al. [144] found that, based on the theoretical results of the Reuss model and the Hashin–Shtrikman equation, for coated beam structures in the ideal state of the model, there existed an optimal coating thickness that could maximize the damping capacity of the coating and substrate when they were not equal. When the modulus difference between the coating and the substrate was increased, it would increase the non-affined degree of the local strain field inside and near the interface and improve the damping performance and strength of the coating. The ratio of the elastic modulus of the coating to the elastic modulus of the substrate has a much greater impact on damping performance than the influence of the components on damping performance. This study provides ideas for improving the performance of composite magnesium materials through composition design by adjusting the elastic modulus and interface parameters between the magnesium matrix and reinforcement phase to regulate the material’s performance. Similarly, in the study by LiMing Yu et al. [145], it was found that due to the interaction between the coating and the substrate under vibration loading, the damping mechanism of the coating sample exhibits a strong nonlinear coupling effect. Proposed by Giuseppe [146], a dynamical model based on a simple beam geometry but taking into account the previously introduced local dissipation mechanism and distributed visco-elastic constraints could also support this point.

Different reinforcing phases will bring different performance improvements to the material. The classical reinforcing phases are graphite, SiC, TiC, Al_2_O_3_ particles, etc. Y. W. Yu et al. [147] prepared graphite-particle-reinforced magnesium matrix composites by stirring casting method, and the composites were extruded at 300 °C, and the extrusion ratio was 12:1. The results showed that graphite particles had an important influence on the damping properties of the materials. When the volume fraction of the ink increases from 0% to 10%, the damping independent of the strain amplitude increases significantly, but it remains almost unchanged when the volume fraction exceeds 10%. Two damping peaks appeared at 150 °C and 350 °C. The former was considered to be caused by movable boundary slip, while the latter was considered to be caused by recrystallization. Y. W. Wu et al. [148] prepared graphite/AZ91 and SiC_p_/graphite/AZ91 composites by stirring casting. They found that the ability of silicon carbide particles to refine grains during solidification was better than that of graphite particles, but graphite particles could promote grain refinement more effectively during hot extrusion. The addition of graphite particles could improve the yield strength and damping performance of the material, but it reduced the ultimate tensile strength. The addition of TiC particles could improve not only the yield strength but also the ultimate tensile strength. Jinhai Gu et al. [149] prepared magnesium matrix composites containing copper-coated and uncoated silicon carbide particles by powder metallurgy and a hot extrusion process and studied their damping properties. The results showed that the damping capacity of pure Mg was higher than that of the composite at low temperatures, but when the temperature reached 75 °C, the damping capacity of the uncoated composite would exceed that of pure Mg. When the temperature was further increased to 250 °C, the damping capacity of the coated composites also exceeded that of pure magnesium.

There are many conditions affecting the damping properties of magnesium matrix composites. The differences in the matrix, reinforcing phase, and the preparation methods used will lead to huge differences in the damping properties of the products, but the overall damping capacity can be better than magnesium alloy and can obtain excellent properties with both high damping properties and mechanical properties. Compared with aluminum matrix composites, the research on the damping properties of magnesium matrix composites started late, and a theoretical system that can solve the problem in a general way has not yet been formed. It requires a lot of scientific research work to design magnesium matrix damping composites with high damping, high strength, and low density.

## 4. High-Damping Porous Magnesium

Porous metal or foam metal materials combined with a metal phase and gas phase possess good development prospects in aerospace, automobiles, construction, and other fields due to their special structure, metal, and pore characteristics, such as low density, large specific surface area, sound absorption, sound insulation, shock absorption, impact energy absorption, and electromagnetic frequency closure [150,151]. When it comes to the damping properties of porous materials, the damping properties of porous metals depend on the porous structure and wall materials [152]. Reference [153] showed that the porous structure of Mg-Al coating on steel could significantly improve the damping performance of materials. The introduction of a porous structure in magnesium could lead to better damping capacity than fully dense magnesium. Since the late 20th century, researchers began to study porous magnesium, including the preparation process, foaming mechanism, mechanical properties analysis, etc. The preparation process is as follows: melt foaming method, seepage casting method, investment casting method, solid–gas eutectic solidification method, powder metallurgy method, and secondary foaming method. The damping capacity of porous magnesium is related to porosity and pore size. The increase in porosity or the decrease in pore size can improve the damping capacity of magnesium alloys. This is due to the inhomogeneity of the microstructure of porous magnesium itself, which puts it in a complex stress–strain state during loading. At the same time, local stress concentration occurs around the hole wall and results in an increase in dislocation density and thus improved damping performance. Zhenkai Xie et al. [154] prepared lotus-shaped porous magnesium with unidirectionally arranged cylindrical pores by unidirectional solidification of molten magnesium under a pressurized hydrogen atmosphere. The damping attenuation coefficient of lotus-shaped porous magnesium was tested by a hammering vibration damping experiment. The results showed that the damping coefficient increases with the increase in porosity; that is, the damping capacity of lotus-shaped magnesium is higher than that of non-porous magnesium. This was due to the existence of pores that stimulated resonance at high frequencies, which can effectively slow down the impact and dissipate energy. Qiuyan, Li et al. [155] developed a new type of porous magnesium alloy. The pores were continuous and curved channels with constant diameter. When the porosity was constant (P = 3%), the loss factor and energy absorption capacity increased with the decrease in pore size. When the pore size was constant (d = 30.0 mm), with the increase in porosity, the dissipation factor increased, but the energy absorption capacity decreased. This pore structure significantly affected the damping capacity and energy absorption capacity of the material.

Wenzhan Huang et al. [156] prepared magnesium alloy/SiC_p_ composite foam by melt foaming process using magnesium carbonate as foaming agent. They found that it had better damping performance than magnesium alloys and pure magnesium. The improvement of damping capacity was mainly due to the micro-slip and micro-plastic deformation caused by micro-cracks between the magnesium alloy and the SiC_p_ interface. Hao Gang et al. [157] prepared porous magnesium with uniform pore distribution by powder metallurgy and measured the damping properties of porous magnesium by internal friction. The results showed that the damping capacity of porous magnesium was improved compared with that of matrix magnesium. This could be explained by the dislocation damping mechanism related to the uneven stress distribution around the pores. A peak value was found in the influence spectrum of temperature on damping. It was speculated to be caused by the dislocation slip caused by thermal activation.

As a new type of lightweight material, porous magnesium alloys have made breakthrough progress in research in recent years. However, there are still many problems, such as the active chemical properties of porous magnesium, easy oxidation and combustion, and increased cost in the protection of samples. At the same time, the control theory and research on the pore structure of closed porous magnesium are relatively lacking, which makes it difficult to control the product quality and limits the promotion of porous magnesium. At present, the research on porous magnesium mainly focuses on millimeter-scale pores, and there are few studies on micron-scale and nano-scale porous magnesium alloys.

## 5. Summary and Outlook

In recent years, with the development of science and technology, the global aerospace, automotive, and other high-tech industries have made great breakthroughs. However, it is also accompanied by the urgent need for vibration and noise-reduction materials. The development of damping magnesium alloys also responds to this. At present, the main ways to enhance the damping of magnesium alloys are to balance the damping performance and mechanical properties by alloying, introducing the second phase, and using a better preparation method.

High-damping materials are widely used in various fields. For magnesium-based high-damping materials, the development of materials, performance improvement, damping mechanism, damping characteristics, and the application of materials are all issues that need to be considered.

Therefore, based on the above research, we have made prospects for the development trend of magnesium alloys, specifically divided into the following points:Compared with magnesium alloys, magnesium matrix composites are more able to meet the requirements of high performance and high damping. The development of damping magnesium alloys has made some progress, and many damping magnesium alloys have been put into use. Just as magnesium matrix composites are added with reinforcing phases, a new damping mechanism is introduced to balance the mechanical properties and damping properties of the alloy. How to introduce a new damping mechanism to make the damping modulating process flexible should be the next development trend of damping magnesium alloys. By adding new strengthening phases and alloying elements, the damping mechanism, besides dislocation damping and interface damping, is introduced to fundamentally solve the contradiction between damping performance and mechanical properties. It is bound to be an important research direction of high-performance and high-damping magnesium alloy materials in the future.Porous magnesium alloy material is also a material with great development potential. Overcoming the inflammable and oxidizable characteristics of magnesium alloy and developing industrial flame-retardant porous magnesium alloys are important issues for its application. At the same time, developing more sophisticated pore size and porosity control technology for porous magnesium alloys is also one of the ways to develop porous magnesium alloys. To further realize the micron and nano research of porous magnesium alloys is an important symbol of the maturity of porous magnesium alloy technology research.To maximize the advantages of vibration and noise reduction of magnesium alloys, it is also a new direction of concern to design product shapes and processing methods that are more conducive to vibration and noise reduction.The damping mechanism of magnesium alloys is still unclear. At present, most of the research experiments on damping mechanisms are only in the stage of modeling and analysis. There are many damping mechanisms, but they are not perfect. Even if the G-L dislocation model is recognized as correct, it only considers the influence of dislocation itself on the damping capacity and has not been subdivided into the specific influence of dislocation type. The influence mechanism of spiral dislocation on damping is still unclear. How to find the optimal dislocation type to further improve the damping is very worthy of study and discussion.Most of the damping magnesium alloys currently used are alloyed to enhance mechanical properties. However, there are still a few technologies that can be used in the preparation of magnesium alloys. Diversifying the preparation of damping magnesium alloys and reducing the preparation cost are important issues for the use of metal magnesium alloying.

## Figures and Tables

**Figure 1 materials-17-01285-f001:**
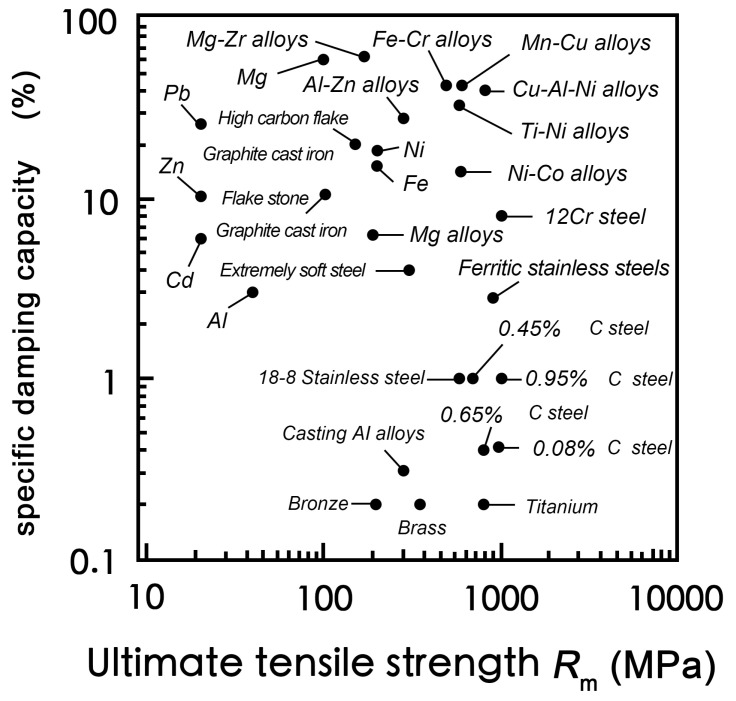
Schematic diagram of specific damping and ultimate tensile strength R_m_ for various damping alloys.

**Figure 2 materials-17-01285-f002:**
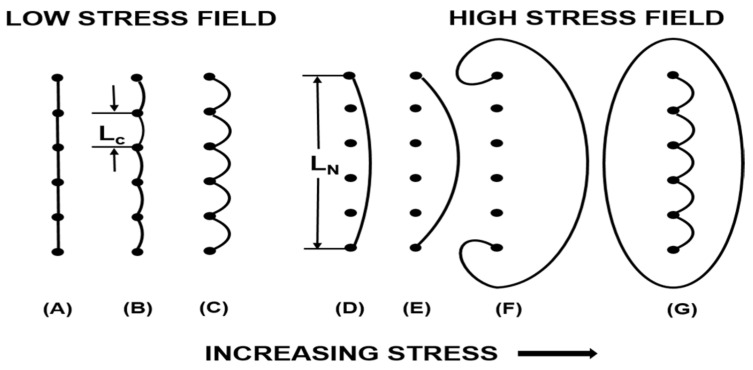
Schematic diagram of G-L dislocation theoretical mode: (**A**–**C**) the process of dislocation movement between two weak pinning points as the stress increases; (**D**,**E**) the process of unpinning movement of a dislocation; (**F**,**G**) the process of dislocation movement between two strong pinning points as the stress increases [42,45].

**Figure 3 materials-17-01285-f003:**
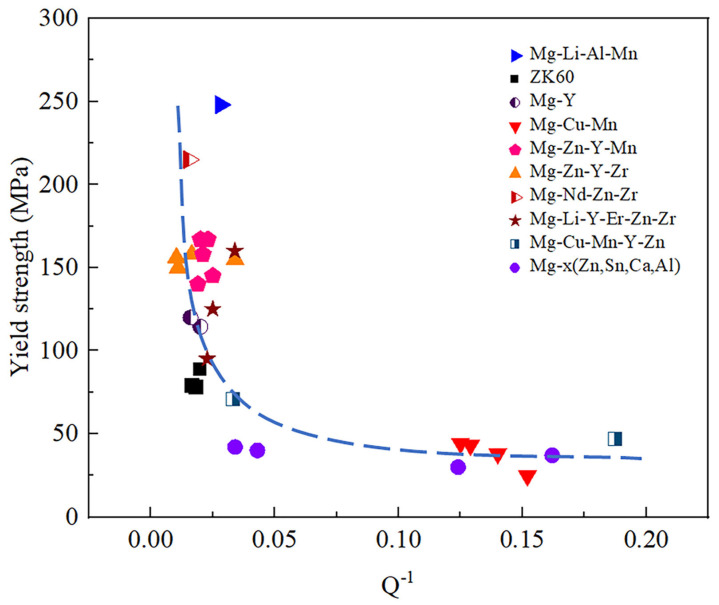
Comparison diagram of damping capacity of different kinds of Mg alloys [48].

**Figure 4 materials-17-01285-f004:**
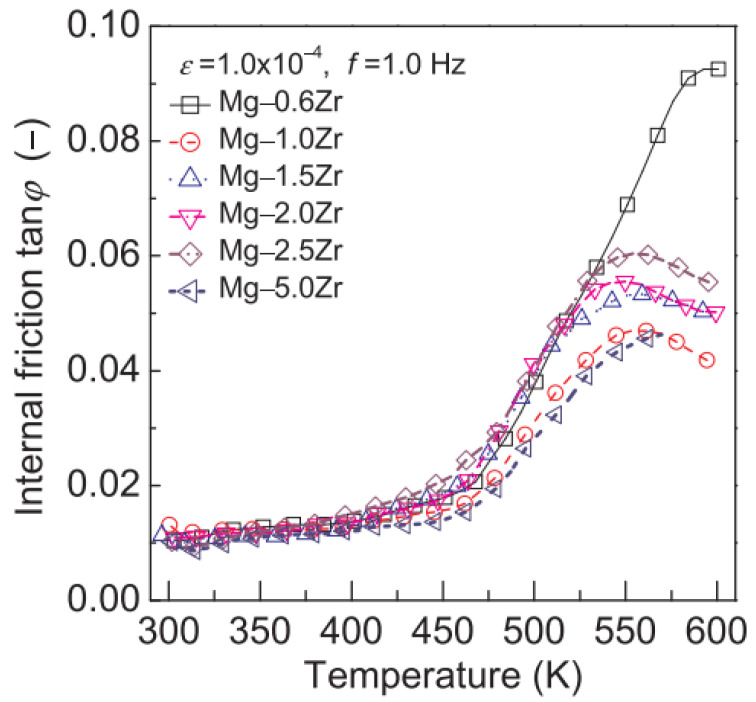
Influence of mass percentage of Zr from 0.6 to 5.0 on T-tan φ plots for six samples of Mg-Zr binary alloys at heating rate of 5 K/min under conditions of strain and frequency, ε = 1.0 × 10^−4^ and f = 1.0 Hz, respectively [54].

**Figure 5 materials-17-01285-f005:**
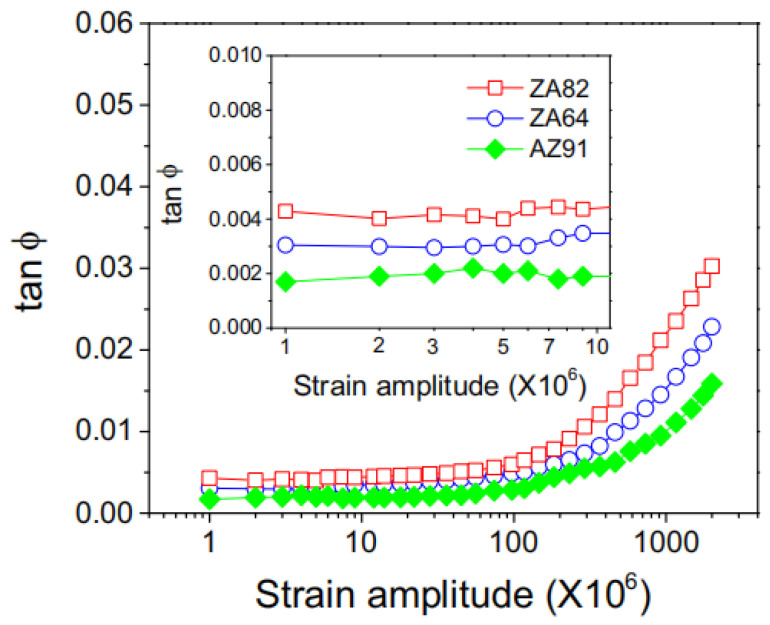
Change in damping capacity (tan φ) with strain amplitude in ZA82, ZA64, and AZ91 alloys [66].

**Figure 6 materials-17-01285-f006:**
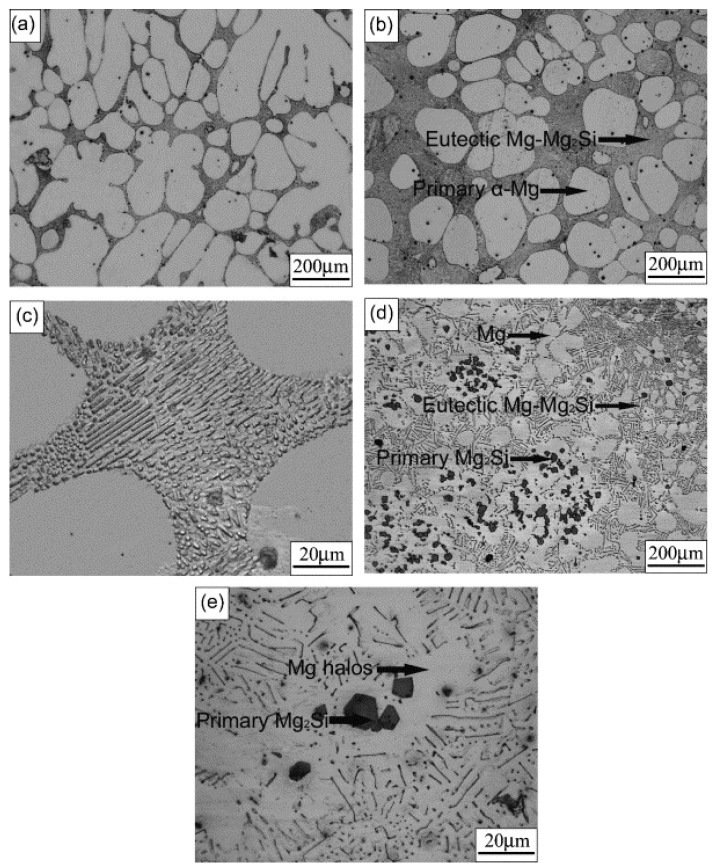
Optical micrographs of (**a**) Mg-0.3 wt.%Si; (**b**) Mg-0.8 wt.%Si; (**c**) higher magnification of Mg-0.8 wt.%Si; (**d**) Mg-2.3 wt.%Si; (**e**) higher magnification of Mg-2.3 wt.%Si [124].

**Table 1 materials-17-01285-t001:** Selected properties for cast magnesium alloys, cast aluminum alloys, titanium alloys, and stainless steel [31].

Properties	Cast Mg Alloys	Cast Al Alloys	Titanium Alloys	Stainless Steel
Density(g/cm^3^) ^a^	1.75–1.87	2.5–2.9 (33%)	4.4–4.8 (61%)	7.6–8.1(77%)
Price (SGD/kg)	4.71–5.17	2.71–2.98	28.7–31.6	7.04–7.75
Melting point (°C)	447–649	475–667	1480–1680	1370–1450
Elastic modulus (GPa)		72–89	110–120	189–210
Specific stiffness	22–27	25–36	23–27	23–28
Yield strength (MPa)	70–215	50–330	750–120	170–1000
Specific strength	37–123	17–132	156–273	21–132
Elongation (% strain)	1–10	0.4–10	5–10	5–70

^a^ Number in parenthesis indicates the percentage difference in density between magnesium and the metal.

**Table 2 materials-17-01285-t002:** Comparison chart of the performances of Mg-Zr alloys.

Alloy Type	Mechanical Properties	Damping Capacity (Q^−1^)	Development Year	Ref.
K1X1-F	σ_B_ = 175 MPa, σ_0.2_ = 60 MPa	/	1960s	[49]
K1-a	/	/	1960s	[49]
Mg-0.6Zr	σ_B_ = 190 MPa, σ_0.2_ = 72 MPa	>0.01	2015	[54]
As-quenched K1	/	0.01724	2011	[57]
Mg-0.6Zr-0.5Zn	σ_b_ = 171 MPa, σ_0.2_ = 77 MPa	>0.01	2019	[50]

**Table 3 materials-17-01285-t003:** Comparison chart of the performances of Mg-Al alloys.

Alloy Type	Mechanical Properties	Damping Capacity (Q^−1^)	Development Year	Ref.
Mg-3Al-xLa	TYS = 160 MPa, EL = 22%	/	2017	[61]
Mg-1Al-xY	/	>0.01	2021	[62]
AZ91-0.5%Ca	/	>0.01	2015	[62]
Stretching-state Mg-1Al	/	>0.01	2010	[64]

**Table 4 materials-17-01285-t004:** Comparison chart of the performances of Mg-Ni alloys.

Alloy Type	Mechanical Properties	Damping Capacity (Q^−1^)	Development Year	Ref.
Mg-0.1%Ni	Improved by a trace of Ni	/	1970s	[32]
Mg-3Ni-xMn	/	Improved by Mn in certain circumstances	2008	[71]
Mg-Ni binary alloy	/	Two damping peaks existed	2005	[72]
Mg-3Ni, Mg-6Ni	/	Several damping plateaus	2009	[73]
Mg-10%Ni, Mg-15%Ni	Comparable to AZ63	30 times that of AZ63	2003	[74]
Mg_95.34_Ni_2_Y_2.66_, Mg_95.34_Zn_1_Ni_1_Y_2.66_	/	>0.01	2022	[78]

**Table 5 materials-17-01285-t005:** Comparison chart of the performances of Mg-Li alloys.

Alloy Type	Mechanical Properties	Damping Capacity (Q^−1^)	Development Year	Ref.
Mg-Li binary alloy	/	/	1910s	[79]
Mg-8.4Li, Mg-9Li	CYS = 192 MPa, UCS = 2312 MPa	Decrease	2022	[14]
Mg-5Li-4Al	high specific strength of 145 kNm/kg	>0.01	2018	[84]
Mg-4Li-3Al-0.3Mn	Its yield strength, tensile strength, and elongation were 248 MPa, 332 MPa, and 14.3%, respectively	0.022	2022	[48]
Mg-Li-Zn	Strength increases, plasticity decreases	>0.01	2022	[91]
Mg-Li binary alloy	/	New damping mechanism	2014	[86]
Mg-13Li-3Al-3Zn	Tensile strength and elongation are 184 MPa and 32%, respectively	>0.01	2023	[92]
Mg-8Li-4Y-2Er-2Zn-0.6Zr	The elastic modulus is 48.9 Gpa; tensile strength is 221 MPa	0.02	2021	[93]
Mg-8Li-6Y-2Zn	The tensile strength is 210 MPa	0.018	2022	[94]

**Table 6 materials-17-01285-t006:** The solid solubility of RE elements in Mg and compounds formed with Mg.

Solubility of RE in Solid Mg and the Possible Compounds Formed with Mg		
RE Element	Solid Solubility (%)	Compound Coexisting with Solid Solutions
Sc	25.9	MgSc
Y	12.0	Mg_24_Y₅
La	0.79	Mg_12_La
Ce	1.6	Mg_12_Ce
Pr	1.7	Mg_12_Pr
Nd	3.6	Mg_12_Nd
Pm	/	/
Sm	5.8	Mg_41_Sm_5_
Eu	/	Mg_17_Eu_2_
Gd	23.5	Mg_5_Gd
Tb	24.0	Mg_5_Tb
Dy	25.8	Mg_24_Dy_5_
Ho	28.0	Mg_24_Ho_5_
Er	32.7	Mg_24_Er_5_
Tm	31.8	Mg_24_Tm_5_
Yb	33	Mg_2_Yb
Lu	41.0	Mg_24_Lu_5_

**Table 7 materials-17-01285-t007:** Comparison chart of the performances of Mg-RE alloys.

Alloy Type	Mechanical Properties	Damping Capacity (Q^−1^)	Development Year	Refs.
Mg-xY	At 325 °C, the yield strength in the extruded state is close to three times that of pure magnesium	Decrease from 0.037 to 0.015 when Y added	2019	[99]
Mg-4Y-2Er-2Zn-0.6Zr	The tensile strength of the rolled alloy reached 362 ± 5 MPa, and the elongation reached 7.8 ± 0.4%	0.018	2022	[101,102]
Mg-Sc	/	Martensitic phase has high damping capacity	2022	[107]
Mg-1.5Gd	/	0.086	2022	[112]
Mg-8.5Gd-5Y-xAl	The tensile strength is 376 MPa, the yield strength is 263 MPa, and the elongation is 13%	0.0132	2021	[113]
Mg-1Gd	Yield strength is 30 MPa	0.16	2022	[114]
Mg-6Gd	Yield strength is 84 MPa	0.05	2022	[114]
Mg-2Ce	Tensile strength and elongation are 184 MPa and 32%, respectively.	0.018	2017	[117]
Mg-1Al-0.5Ce	Tensile yield strength is 169 MPa	0.035	2017	[118]

**Table 8 materials-17-01285-t008:** Comparison chart of the performances of Mg-Cu-Mn alloys.

Alloy Type	Mechanical Properties	Damping Capacity (Q^−1^)	Development Year	Ref.
CM31	tensile strength is 290 MPa	specific damping value is 60%	2003	[10]
Mg-2.5Cu-0.8Mn	yield strength is 43 MPa, ultimate tensile strength is 131 MPa, elongation is 6.81%,	>0.01	2008	[131]
Mg-3Cu-1Mn-1Y-2Zn	the tensile strength and yield strength of the alloy were 71 MPa and 47 MPa, respectively	>0.01	2010	[134]
Mg-Cu-Mn-Zn-Y	/	>0.01, and a new frequency-dependent damping mechanism was found	2012	[135]

## Data Availability

Not applicable.

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
