# Peer review of "Research Progress and the Prospect of Damping Magnesium Alloys"

_materials, 2024, doi:10.3390/ma17061285_

Round 1

Reviewer 1 Report

Comments and Suggestions for Authors

This review states that the objective of this work is to present the progress in the research of damping magnesium alloys with integrated structure and function, and predict their development trends.

Intention and idea are important when it comes to magnesium alloys; However, the content and format of the present work is deficient for a review presentation. The work is rather a compendium of the different magnesium damping alloys without any chronology although in the title they mention that it is a “research advance.”

They also do not conclude with any perspective on magnesium damping alloys. The authors should present by separately  the mechanical properties of damping magnesium alloys or do more research on the effect of dislocation on damping characteristics or consider different trends, maybe this can be more attractive. Additionally, they should consider including more images or schemes.

Author Response

See the attached reply report.

Reviewer 2 Report

Comments and Suggestions for Authors

See the attached review report.

Comments on the Quality of English Language

Several small English Language mistakes are present throughout the text, which must be carefully corrected by the Authors.

Author Response

See the attached reply report.

Round 2

Reviewer 1 Report

Comments and Suggestions for Authors

The authors have improved the manuscript giving greater relevance to their investigation.